# General Stochastic Networks for Classification

**Matthias Zöhrer and Franz Pernkopf**
Signal Processing and Speech Communication Laboratory
Graz University of Technology
matthias.zoehrer@tugraz.at, pernkopf@tugraz.at

## Abstract

We extend generative stochastic networks to supervised learning of representations. In particular, we introduce a hybrid training objective considering a generative and discriminative cost function governed by a trade-off parameter $\lambda$. We use a new variant of network training involving noise injection, i.e. *walkback* training, to jointly optimize multiple network layers. Neither additional regularization constraints, such as $\ell 1$, $\ell 2$ norms or dropout variants, nor pooling- or convolutional layers were added. Nevertheless, we are able to obtain state-of-the-art performance on the MNIST dataset, without using permutation invariant digits and outperform baseline models on sub-variants of the MNIST and *rectangles* dataset significantly.

## 1 Introduction

Since 2006 there has been a boost in machine learning due to improvements in the field of unsupervised learning of representations. Most accomplishments originate from variants of restricted Boltzmann machines (RBMs) [1], auto-encoders (AE) [2, 3] and sparse-coding [4, 5, 6]. *Deep* models in representation learning, also obtain impressive results in supervised learning problems, such as speech recognition, e.g. [7, 8, 9] and computer vision tasks [10].

If no a-priori knowledge is modeled in the architecture, cf. convolutional layers or pooling layers [11], generatively pre-trained networks are among the best when applied to supervised learning tasks [12]. Usually, a generative representation is obtained through a greedy-layerwise training procedure called contrastive divergence (CD) [1]. In this case, the network layer learns the representation from the layer below by treating the latter as static input. Despite of the impressive results achieved with CD, we identify two (minor) drawbacks when used for supervised learning: Firstly, after obtaining a representation by pre-training a network, a new discriminative model is initialized with the trained weights, splitting the training into two separate models. This seems to be neither biologically plausible, nor optimal when it comes to optimization, as carefully designed early stopping criteria have to be implemented to prevent over- or under-fitting. Secondly, generative and discriminative objectives might influence each other beneficially when combined during training. CD does not take this into account.

In this work, we introduce a new training procedure for supervised learning of representations. In particular we define a hybrid training objective for general stochastic networks (GSN), dividing the cost function into a generative and discriminative part, controlled by a trade-off parameter $\lambda$. It turns out that by annealing $\lambda$, when solving this unconstrained non-convex multi-objective optimization problem, we do not suffer from the shortcomings described above. We are able to obtain state-of-the-art performance on the MNIST [13] dataset, without using permutation invariant digits and significantly outperform baseline models on sub-variants of the MNIST and *rectangle* database [14].

Our approach is related to the generative-discriminative training approach of RBMs [15]. However a different model and a new variant of network training involving noise injection, i.e. *walkback* training [16, 17], is used to jointly optimize multiple network layers. Most notably, we did not

apply any additional regularization constraints, such as $\ell1$, $\ell2$ norms or dropout variants [12], [18], unlocking further potential for possible optimizations. The model can be extended to learn multiple tasks at the same time using jointly trained weights and by introducing multiple objectives. This might also open a new prospect in the field of transfer learning [19] and multi-task learning [20] beyond classification.

This paper is organized as follows: Section 2 presents mathematical background material i.e. the GSN and a hybrid learning criterion. In Section 3 we empirically study the influence of hyper parameters of GSNs and present experimental results. Section 4 concludes the paper and provides a perspective on future work.

## 2 General Stochastic Networks

Recently, a new supervised learning algorithm called *walkback* training for generalized auto-encoders (GAE) was introduced [16]. A follow-up study [17] defined a new network model – generative stochastic networks, extending the idea of *walkback* training to multiple layers. When applied to image reconstruction, they were able to outperform various baseline systems, due to its ability to learn multi-modal representations [17, 21]. In this paper, we extend the work of [17]. First, we provide mathematical background material for generative stochastic networks. Then, we introduce modifications to make the model suitable for supervised learning. In particular we present a hybrid training objective, dividing the cost into a generative and discriminative part. This paves the way for any multi-objective learning of GSNs. We also introduce a new terminology, i.e. general stochastic networks, a model class including generative-, discriminative- and hybrid stochastic network variants.

**General Stochastic Networks for Unsupervised Learning**

Restricted Boltzmann machines (RBM) [22] and denoising autoencoders (DAE) [3] share the following commonality; The input distribution $P(X)$ is sampled to convergence in a Markov chain. In the case of the DAE, the transition operator first samples the hidden state $H_t$ from a corruption distribution $C(H|X)$, and generates a reconstruction from the parametrized model, i.e the density $P_{\theta_2}(X|H)$.

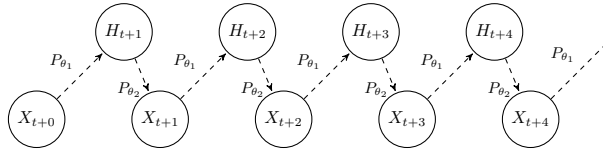

Figure 1: *DAE Markov chain.*

The resulting DAE Markov chain, shown in Figure 1, is defined as

$$H_{t+1} \sim P_{\theta_1}(H|X_{t+0}) \text{ and } X_{t+1} \sim P_{\theta_2}(X|H_{t+1}), \tag{1}$$

where $X_{t+0}$ is the input sample $X$, fed into the chain at time step 0 and $X_{t+1}$ is the reconstruction of $X$ at time step 1. In the case of a GSN, an additional dependency between the latent variables $H_t$ over time is introduced to the network graph. The GSN Markov chain is defined as follows:

$$H_{t+1} \sim P_{\theta_1}(H|H_{t+0}, X_{t+0}) \text{ and } X_{t+1} \sim P_{\theta_2}(X|H_{t+1}). \tag{2}$$

Figure 2 shows the corresponding network graph.

This chain can be expressed with deterministic functions of random variables $f_\theta \supseteq \{\hat{f}_\theta, \check{f}_\theta\}$. In particular, the density $f_\theta$ is used to model $H_{t+1} = f_\theta(X_{t+0}, Z_{t+0}, H_{t+0})$, specified for some independent noise source $Z_{t+0}$, with the condition that $X_{t+0}$ cannot be recovered exactly from $H_{t+1}$.

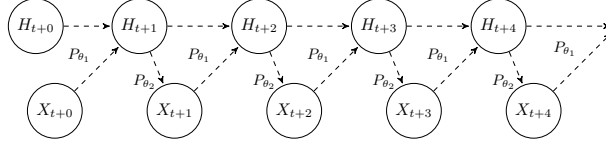

Figure 2: *GSN Markov chain.*

We introduce $\hat{f}_\theta^i$ as a back-probable stochastic non-linearity of the form $\hat{f}_\theta^i = \eta_{out} + g(\eta_{in} + \hat{a}_i)$ with noise processes $Z_t \supseteq \{\eta_{in}, \eta_{out}\}$ for layer $i$. The variable $\hat{a}^i$ is the activation for unit $i$, where $\hat{a}^i = W^i I_t^i + b^i$ with a weight matrix $W^i$ and bias $b^i$, representing the parametric distribution. It is embedded in a non-linear activation function $g$. The input $I_t^i$ is either the realization $x_t^i$ of observed sample $X_t^i$ or the hidden realization $h_t^i$ of $H_t^i$. In general, $\hat{f}_\theta^i(I_t^i)$ specifies an upward path in a GSN for a specific layer $i$. In the case of $X_{t+1}^i = \check{f}_\theta^i(Z_{t+0}, H_{t+1})$ we define $\check{f}_\theta^i(H_t^i) = \eta_{out} + g(\eta_{in} + \check{a}_i)$ as a downward path in the network i.e. $\check{a}^i = (W^i)^T H_t^i + b^i$, using the transpose of the weight matrix $W^i$ and the bias $b^i$. This formulation allows to directly back-propagate the reconstruction log-likelihood $P(X|H)$ for all parameters $\theta \supseteq \{W^0, ..., W^d, b^0, ..., b^d\}$ where $d$ is the number of hidden layers. In Figure 2 the GSN includes a simple hidden layer. This can be extended to multiple hidden layers requiring multiple deterministic functions of random variables $f_\theta \in \{\hat{f}_\theta^0, ..., \hat{f}_\theta^d, \check{f}_\theta^0, ...\check{f}_\theta^d\}$.

Figure 3 visualizes the Markov chain for a multi-layer GSN, inspired by the unfolded computational graph of a deep Boltzmann machine Gibbs sampling process.

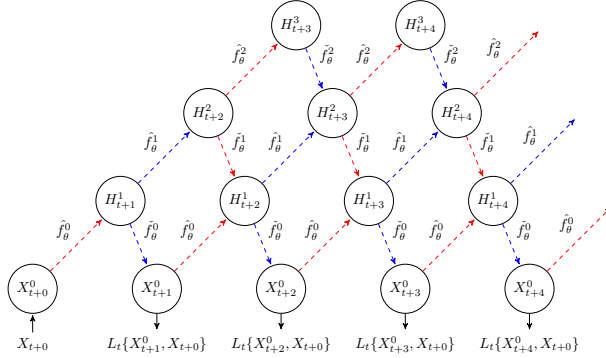

Figure 3: *GSN Markov chain with multiple layers and backprop-able stochastic units.*

In the training case, alternatively even or odd layers are updated at the same time. The information is propagated both upwards and downwards for $K$ steps allowing the network to build higher order representations. An example for this update process is given in Figure 3. In the even update (marked in *red*) $H_{t+1}^1 = \hat{f}_\theta^0(X_{t+0}^0)$. In the odd update (marked in *blue*) $X_{t+1}^0 = \check{f}_\theta^0(H_{t+1}^1)$ and $H_{t+2}^2 = \hat{f}_\theta^1(H_{t+1}^1)$ for $k = 0$. In the case of $k = 1$, $H_{t+2}^1 = \hat{f}_\theta^0(X_{t+1}^0) + \check{f}_\theta^1(H_{t+2}^2)$ and $H_{t+3}^3 = \hat{f}_\theta^2(H_{t+2}^2)$ in the even update and $X_{t+2}^0 = \check{f}_\theta^0(H_{t+2}^1)$ and $H_{t+3}^2 = \hat{f}_\theta^1(H_{t+2}^1) + \check{f}_\theta^2(H_{t+3}^3)$ in the odd update. In case of $k = 2$, $H_{t+3}^1 = \hat{f}_\theta^0(X_{t+2}^0) + \check{f}_\theta^1(H_{t+3}^2)$ and $H_{t+4}^3 = \hat{f}_\theta^2(H_{t+3}^2)$ in the even update and $X_{t+3}^0 = \check{f}_\theta^0(H_{t+3}^1)$ and $H_{t+4}^2 = \hat{f}_\theta^1(H_{t+3}^1) + \check{f}_\theta^2(H_{t+4}^3)$ in the odd update.

The cost function of a generative GSN can be written as:

$$C = \sum_{k=1}^{K} L_t\{X_{t+k}^0, X_{t+0}\},$$
(3)

$L_t$ is a specific loss-function such as the mean squared error (MSE) at time step $t$. In general any arbitrary loss function could be used (as long as they can be seen as a log-likelihood) [16]. $X^0_{t+k}$ is the reconstruction of the input $X^0_{t+0}$ at layer 0 after $k$ steps. Optimizing the loss function by building the sum over the costs of multiple corrupted reconstructions is called *walkback* training [16, 17]. This form of network training leads to a significant performance boost when used for input reconstruction. The network is able to handle multi-modal input representations and is therefore considerably more favorable than standard generative models [16].

### General Stochastic Networks for Supervised Learning

In order to make a GSN suitable for a supervised learning task we introduce the output $Y$ to the network graph. In this case $L = \log P(X) + \log P(Y|X)$. Although the target $Y$ is not fed into the network, it is introduced as an additional cost term. The layer update-process stays the same.

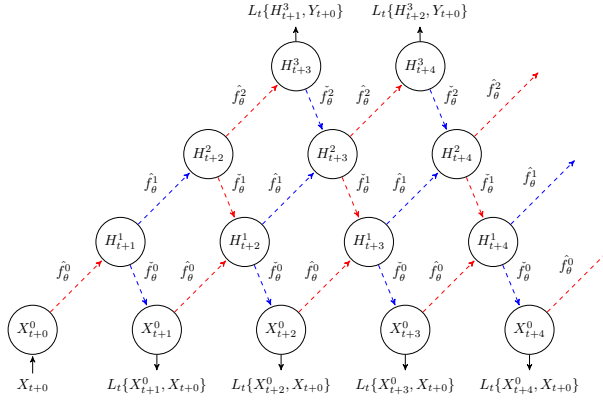

Figure 4: *GSN Markov chain for input $X_{t+0}$ and target $Y_{t+0}$ with backprop-able stochastic units.*

We define the following cost function for a 3-layer GSN:

$$C = \underbrace{\frac{\lambda}{K} \sum_{k=1}^{K} L_t\{X_{t+k}, X_{t+0}\}}_{generative} + \underbrace{\frac{1-\lambda}{K-d+1} \sum_{k=d}^{K} L_t\{H^3_{t+k}, Y_{t+0}\}}_{discriminative} \tag{4}$$

This is a non-convex multi-objective optimization problem, where $\lambda$ weights the generative and discriminative part of $C$. The parameter $d$ specifies the number of network layers i.e. depth of the network. Scaling the mean loss in (4) is not mandatory, but allows to equally balance both loss terms with $\lambda = 0.5$ for input $X_{t+0}$ and target $Y_{t+0}$ scaled to the same range. Again Figure 4 shows the corresponding network graph for supervised learning with *red* and *blue* edges denoting the even and odd network updates.

In general the hybrid objective optimization criterion is not restricted to $\langle X, Y \rangle$, as additional input and output terms could be introduced to the network. This setup might be useful for transfer-learning [19] or multi-task scenarios [20], which is not discussed in this paper.

## 3   Experimental Results

In order to evaluate the capabilities of GSNs for supervised learning, we studied MNIST digits [13], variants of MNIST digits [14] and the *rectangle* datasets [14]. The first database consists of 60.000 labeled training and 10.000 labeled test images of handwritten digits. The second dataset includes 6 variants of MNIST digits, i.e. { *mnist-basic*, *mnist-rot*, *mnist-back-rand*, *mnist-back-image*, *mnist-rot-back-image* }, with additional factors of variation added to the original data. Each variant includes 10.000 labeled training, 2000 labeled validation, and 50.000 labeled test images. The third dataset involves two subsets, i.e. { *rectangle*, *rectangle-image* }. The dataset *rectangle* consists of

1000 labeled training, 200 labeled validation, and 50.000 labeled test images. The dataset *rectangle-image* includes 10.000 labeled train, 2000 labeled validation and 50.000 labeled test images.

In a first experiment we focused on the multi-objective optimization problem defined in (4). Next we evaluated the number of *walkback* steps in a GSN, necessary for convergence. In a third experiment we analyzed the influence of different Gaussian noise settings during *walkback* training, improving the generalization capabilities of the network. Finally we summarize classification results for all datasets and compare to baseline systems [14].

### 3.1 Multi-Objective Optimization in a Hybrid Learning Setup

In order to solve the non-convex multi-objective optimization problem, variants of stochastic gradient descent (SGD) can be used. We applied a search over fixed $\lambda$ values on all problems. Furthermore, we show that the use of an annealed $\lambda$ factor, during training works best in practice.

In all experiments a three layer GSN, i.e. GSN-3, with 2000 neurons in each layer, randomly initialized with small Gaussian noise, i.e. $0.01 \cdot \mathcal{N}(0,1)$, and an MSE loss function for both inputs and targets was used. Regarding optimization we applied SGD with a learning rate $\eta = 0.1$, a momentum term of 0.9 and a multiplicative annealing factor $\eta_{n+1} = \eta_n \cdot 0.99$ per epoch $n$ for the learning rate. A rectifier unit [23] was chosen as activation function. Following the ideas of [24] no explicit sampling was applied at the input and output layer. In the test case the zero-one loss was computed averaging the network's output over k *walkback* steps.

**Analysis of the Hybrid Learning Parameter $\lambda$**

Concerning the influence of the trade-off parameter $\lambda$, we tested fixed $\lambda$ values in the range $\lambda \in \{0.01, 0.1, 0.2, ..., 0.9, 0.99\}$, where low values emphasize the discriminative part in the objective and vice versa. *Walkback* training with $K = 6$ steps using zero-mean pre- and post-activation Gaussian noise with zero mean and variance $\sigma = 0.1$ was performed for 500 training epochs. In a more dynamic scenario $\lambda_{n=1} = 1$ was annealed by $\lambda_{n+1} = \lambda_n \cdot \tau$ to reach $\lambda_{n=500} \in \{0.01, 0.1, 0.2, ..., 0.9, 0.99\}$ within 500 epochs, simulating generative pre-training to a certain extend.

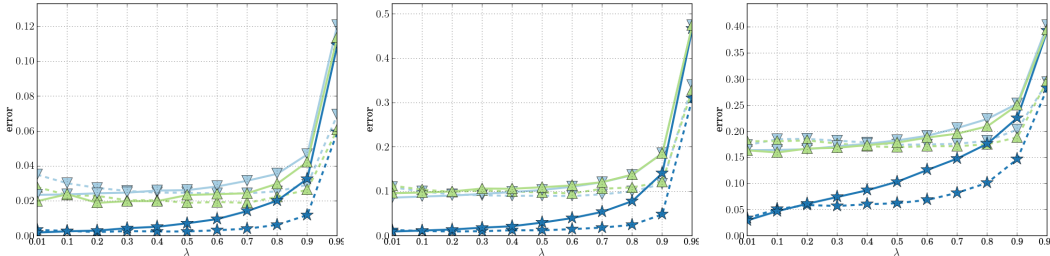

Figure 5: Influence of dynamic and static $\lambda$ on MNIST variants basic (*left*), rotated (*middle*) and background (*right*) where $\star$ denotes the *training-*, $\triangle$ the *validation-* and $\triangledown$ the *test*-set. The dashed line denotes the *static* setup, the bold line the *dynamic* setup.

Figure 5 compares the results of both GSNs, using static and dynamic $\lambda$ setups on the MNIST variants *basic*, *rotated* and *background*. The use of a dynamic i.e. annealed $\lambda_{n=500} = 0.01$, achieved the best validation and test error in all experiments. In this case, more attention was given to the generative proportion $P(X)$ of the objective (4) in the early stage of training. After approximately 400 epochs discriminative training i.e. *fine-tuning*, dominates. This setup is closely related to DBN training, where emphasis is on optimizing $P(X)$ at the beginning of the optimization, whereas $P(Y|X)$ is important at the last stages. In case of the GSN, the annealed $\lambda$ achieves a more smooth transition by shifting the weight in the optimization criterion from $P(X)$ to $P(Y|X)$ within one model.

**Analysis of Walkback Steps $K$**

In a next experiment we tested the influence of $K$ *walkback* steps for GSNs. Figure 6 shows the results for different GSNs, trained with $K \in \{6, 7, 8, 9, 10\}$ *walkback* steps and annealed $\lambda$ with $\tau = 0.99$. In all cases the information was at least propagated once up and once downwards in the $d = 3$ layer network using fixed Gaussian pre- and post-activation noise with $\mu = 0$ and $\sigma = 0.1$.

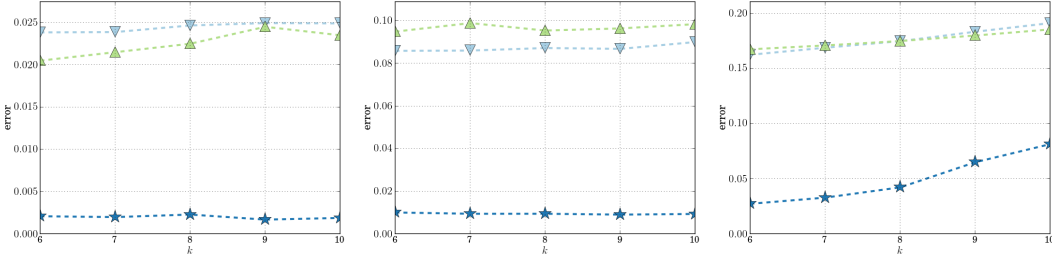

Figure 6: Evaluating the number of *walkback* steps on MNIST variants basic (*left*), rotated (*middle*) and background (*right*) where $\star$ denotes the *training-*, $\triangle$ the *validation-* and $\triangledown$ the *test*-set.

Figure 6 shows that increasing the *walkback* steps, does not improve the generalization capabilities of the used GSNs. The setup $K = 2 \cdot d$ is sufficient for convergence and achieves the best validation and test result in all experiments.

**Analysis of Pre- and Post-Activation Noise**

Injecting noise during the training process of GSNs serves as a regularizer and improves the generalization capabilities of the model [17]. In this experiment the influence of Gaussian pre- and post-activation noise with $\mu = 0$ and $\sigma \in \{0.05, 0.1, 0.15, 0.2, 0.25, 0.3\}$ and deactivated noise during training, was tested on a GSN-3 trained for $K = 6$ *walkback* steps. The trade-off factor $\lambda$ was annealed with $\tau = 0.99$. Figure 7 summarizes the results of the different GSNs for the MNIST variants *basic*, *rotated* and *background*. Setting $\sigma = 0.1$ achieved the best overall result on the validation- and test-set for all three experiments. In all other cases the GSNs either over- or underfitted the data.

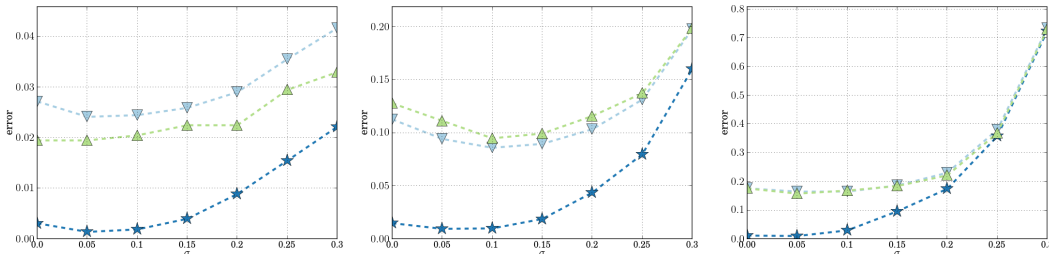

Figure 7: Evaluating noise injections during training on MNIST variants basic (*left*), rotated (*middle*) and background (*right*) where $\star$ denotes the *training-*, $\triangle$ the *validation-* and $\triangledown$ the *test*-set.

## 3.2 MNIST results

Table 1 presents the average classification error of three runs of all MNIST variation datasets obtained by a GSN-3, using fixed Gaussian pre- and post-activation noise with $\mu = 0$, $\sigma = 0.1$ and $K = 6$ *walkback* steps. The hybrid learning parameter $\lambda$ was annealed with $\tau = 0.99$ and $\lambda_{n=1} = 1$. A small grid test was performed in the range of $N \times d$ with $N \in \{1000, 2000, 3000\}$ neurons per layer for $d \in \{1, 2, 3\}$ layers to find the optimal network configuration.

| Dataset | SVMrbf | SVMpoly | NNet | DBN-1 | SAA-3 | DBN-3 | GSN-3 |
|---|---|---|---|---|---|---|---|
| mnist-basic | 3.03 ±0.15 | 3.69 ±0.17 | 4.69 ±0.19 | 3.94 ±0.17 | 3.46 ±0.16 | 3.11 ±0.15 | **2.40 ±0.04** |
| mnist-rot* | 11.11 ±0.28 | 15.42 ±0.32 | 18.11 ±0.34 | 10.30 ±0.27 | 10.30 ±0.27 | 14.69 ±0.31 | **8.66 ±0.08** |
| mnist-back-rand | 14.58 ±0.31 | 16.62 ±0.33 | 20.04 ±0.35 | 9.80 ±0.26 | 11.28 ±0.28 | **6.73 ±0.22** | 9.38 ±0.03 |
| mnist-back-image | 22.61 ±0.37 | 24.01 ±0.37 | 27.41 ±0.39 | 16.15 ±0.32 | 23.00 ±0.37 | 16.31 ±0.32 | **16.04 ±0.04** |
| mnist-rot-back-image* | 55.18 ±0.44 | 56.41 ±0.43 | 62.16 ±0.43 | 47.39 ±0.44 | 51.93 ±0.44 | 52.21 ±0.44 | **43.86 ±0.05** |
| rectangles | 2.15 ±0.13 | 2.15 ±0.13 | 7.16 ±0.23 | 4.71 ±0.19 | 2.41 ±0.13 | 2.60 ±0.14 | **2.04 ±0.04** |
| rectangles-image | 24.04 ±0.37 | 24.05 ±0.37 | 33.20 ±0.41 | 23.69 ±0.37 | 24.05 ±0.37 | 22.50 ±0.37 | **22.10 ±0.03** |

Table 1: MNIST *variations* and *recangle* results [14]; For datasets marked by (*) updated results are shown [25].

Table 1 shows that a three layer GSN clearly outperforms all other models, except for the MNIST *random-background* dataset. In particular, when comparing the GSN-3 to the radial basis function support vector machine (SVMrbf), i.e. the second best model on MNIST *basic*, the GSN-3 achieved an relative improvement of 20.79% on the test set. On the MNIST *rotated* dataset the GSN-3 was able to beat the second best model i.e. DBN-1, by 15.92% on the test set. On the MNIST *rotated-background* there is an relative improvement of 7.25% on the test set between the second best model, i.e. DBN-1, and the GSN-3. All results are statistically significant. Regarding the number of model parameters, although we cannot directly compare the models in terms of network parameters, it is worth to mention that a far smaller grid test was used to generate the results for all GSNs, cf. [14]. When comparing the classification error of the GSN-3 trained without noise, obtained in the previous experiments (7) with Table 1, the GSN-3 achieved the test error of 2.72% on the MNIST variant *basic*, outperforming all other models on this task. On the MNIST variant *rotated*, the GSN-3 also outperformed the DBN-3, obtaining a test error of 11.2%. This indicates that not only the Gaussian regularizer in the *walkback* training improves the generalization capabilities of the network, but also the hybrid training criterion of the GSN.

Table 2 lists the results for the MNIST dataset without additional affine transformations applied to the data i.e. permutation invariant digits. A three layer GSN achieved the state-of-the-art test error of 0.80%.

| Network | Result |
|---|---|
| Rectifier MLP + dropout [12] | 1.05% |
| DBM [26] | 0.95% |
| Maxout MLP + dropout [27] | 0.94% |
| MP-DBM [28] | 0.91% |
| Deep Convex Network [29] | 0.83% |
| Manifold Tangent Classifier [30] | 0.81% |
| DBM + dropout [12] | **0.79%** |
| GSN-3 | 0.80% |

Table 2: MNIST results.

It might be worth noting that in addition to the noise process in *walkback* training, no other regularizers, such as $\ell 1$, $\ell 2$ norms and dropout variants [12], [18] were used in the GSNs. In general $\leq 800$ training epochs with early-stopping are necessary for GSN training.

All simulations[1] were executed on a GPU with the help of the mathematical expression compiler Theano [31].

## 4   Conclusions and Future Work

We have extended GSNs for classification problems. In particular we defined an hybrid multi-objective training criterion for GSNs, dividing the cost function into a generative and discriminative part. This renders the need for generative pre-training unnecessary. We analyzed the influence of the objective's trade-off parameter $\lambda$ empirically, showing that by annealing $\lambda$ we outperform a static choice of $\lambda$. Furthermore, we discussed effects of noise injections and sampling steps during *walkback* training. As a conservative starting point we restricted the model to use only rectifier units. Neither additional regularization constraints, such as $\ell 1$, $\ell 2$ norms or dropout variants [12], [18], nor pooling- [11, 32] or convolutional layers [11] were added. Nevertheless, the GSN was able to outperform various baseline systems, in particular a deep belief network (DBN), a multi layer perceptron (MLP), a support vector machine (SVM) and a stacked auto-associator (SSA), on variants of the MNIST dataset. Furthermore, we also achieved state-of-the-art performance on the original MNIST dataset without permutation invariant digits. The model not only converges faster in terms of training iterations, but also show better generalization behavior in most cases. Our approach opens a wide field of new applications for GSNs. In future research we explore adaptive noise injection methods for GSNs and non-convex multi-objective optimization strategies.

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
