[Reviews · NeurIPS 2014]

Submitted by Assigned_Reviewer_5

This paper described a supervised version on a recently proposed unsupervised generative stochastic networks. The trick is to put a loss on the difference between the label highest-level hidden units, and jointly train on the supervised and unsupervised tasks. The user did extensive experiments on hyper- parameter analysis and achieved near state-of-art performance on MNIST dataset.

The paper is very well written and is easy to follow. The background of GSN and the proposed method have been made very clear. The experiments are sufficient and well analyzed. The idea of the paper is somewhat natural because similar tricks have been used on other generative methods like LDA and RBM. But the experiments and the analysis made it a good paper.

Questions:
1. should it be "Generative" or "General" Stochastic Network?
2. Is there a reason why the walk-back be at least 2*d?
Summary: A well written paper with good experiment results and good analysis.

Submitted by Assigned_Reviewer_6

This paper extends the General Stochastic Network (GSN) models to perform simultaneous generative and discriminative tasks, via a hybrid training objective that balances a data reconstruction and label prediction. The model is trained using the recently introduced walkback algorithm, which yields a generative model by first injecting noise and then learning to remove it, and provides an alternative to other regularization techniques. The model is trained on several variants of MNIST, showing improvement with respect to previous methods.

The paper has two main contributions. (i) extension of GSNs to perform simultaneous discriminative and generative task, (ii) introduce a hybrid cost function, with a trade-off parameter that controls the balance between discriminative and generative terms and an annealing scheme. I think that each contribution, taken individually, is relatively minor and incremental, but the overall model is interesting, with potential to drive further improvements, and shown to be effective in several tasks.

Pros:

- Good numerical results: state-of-the-art results on MNIST permutation invariant ( If I understood correctly, the performance of the network on that dataset would have been the same if the pixels were shuffled, hence it is not using the pixel geometry - if that is the case, I do not understand the terminology 'without permutation invariant digits' ).

- Interesting alternative to the classical unsupervised pretraining + discriminative fine-tuning paradigm. Although the annealing scheme effectively performs something similar in spirit, the transition is smoother and the framework is easier to study. The reviewer is however not expert enough to say whether this is a novelty.

- Potential for improvement: As the authors mention, the model seems robust to hyperparameters (number of walkback steps and noise level) and prone to generalization to more sophisticated tasks.

Cons/Remarks:

- Somehow the paper mixes two different aspects that might confuse the reader: (i) how to combine generative with discriminative tasks, and (ii) how to obtain efficient generative models. The paper considers the GSN framework, which is a valid choice, but I would have liked to see a more clear comparison (perhaps a section?) where this model is compared with traditional (noise-free) autoencoders which use a reconstruction term and a discriminative term. The authors mention in the numerical experiments section that the GSN model with no noise also performed well in experiments. Is that equivalent to a standard transforming autoencoder enriched with a discriminative cost?

- Additive interactions in the Markov chains. Figures 2 and 3 show the basic architecture of the GSN, which generalizes the Denoising Autoencoder Markov chain. In this extended model, hidden random variables are modeled as sums of transition functions coming from different sources. I would have expected the influence of different sources to have a nonlinear effect in the state of hidden variables, such as multiplicative interactions.

- Is the Gaussian additive noise (pre and post activation) sufficient to achieve multimodality? Could the authors describe how this mechanism scales to high-dimensional data (with potentially the number of 'modes' scaling exponentially with the dimension).
Section 3.1 describes an analysis of the noise sigma, but sigma is not defined previously (I assumed it is the standard deviation of eta_out and eta_in). Is this noise isotropic? why?

- Interpretation of section 3.1. The analysis of the role of walkback steps and the noise level is critical for the understanding of the model, and I wished the authors would give further, deeper insight on the results depicted in Figures 6 and 7. Why does the performance depend so little on K? Has the chain already converged? And also, from Figure 7 is seems as if the noise-free version performs nearly as well as the carefully chosen noisy version. Is this because the noise distributions are too simplistic (Isotropic gaussian, with same variance across the network?), or is there a more fundamental reason?

Summary: Overall, I liked the paper. I think it is important to understand how to combine unsupervised and supervised learning, and this paper presents a valid and theoretically interesting model that goes into that direction, and provides good numerical performance (albeit not yet on challenging real-world datasets). The only concern is novelty with respect to [13] and [17], which is given by the hybrid generative-discriminative cost function.

Submitted by Assigned_Reviewer_15

The paper is proposing an interesting variant of multilayer GSN "walkback" training of Bengio, et al., just published in ICML 2014. The idea is to add a penalty for predicting labels at the top of the network. I think that GSNs are a very promising new way of building generative models. The experimental results for the discriminative version is not very compelling

Significance: I think that GSNs are potentially very impactful: they are a new approach to unsupervised learning, without all of the pain of marginalizing or coping with multimodal posteriors. This paper shows that GSNs can be trained discriminatively, which could add to the impact of GSNs. The incremental change in the GSN model is quite small: a simple addition of a penalty at the top of the network.

Quality: My main concern with this paper is that the experimental results are only on a set of variants of MNIST (plus rectangles) that have not spread much beyond the original authors of the paper reference [15]. The results on all of those data sets are very good -- substantially better than any othertechnique tried. My sense is that a few more weeks of work can show that discriminative GSNs can be a very useful model on real-world data, such as CIFAR or ImageNet. Perhaps the authors should consider submitting to ICLR or ICML with some great results?

What's "back-probable" ? Is it a typo for "back-propable"?
Summary: An interesting discriminative version of a GSN. I wish that the experiments went beyond MNIST variants and rectangles.
Author Feedback
Author rebuttal: Reviewer_15

Thank you very much for your comments and suggestions. We are aware of the relatively limited number of datasets. We aim to extend the experimental section by CIFAR10, SVHN, STL10. We wanted to keep the model complexity as small as possible, making the paper easy to follow and reproducible. Therefore we did not add convolutional layers or pooling operations to the model. Running tests on ImageNet might not lead to competitive results without these additional structures (cf. Krivitsky,Hinton).

Reviewer 5, Reviewer 6

Thank you much for your comments and suggestions. Regarding the impact of the paper, the presented model is build out of multiple small contributions, if taken individually -- relatively minor, but when combined together leading to something interesting and powerful (shown to be effective in several tasks). We proposed a new network model capable to jointly optimize multiple layers of a neural network with a hybrid generative-discriminative training objective, i.e. an interesting alternative to the classical unsupervised pre-training + discriminative fine-tuning paradigm. To our mind this has not been done before and opens up new perspectives for follow-up studies. It might have impact on the community.

Regarding the terminology General Stochastic Network (GSN): GSNs has been initially defined as generative models (1). This paper presents a new hybrid GSN (2). In order to differentiate between (1) and (2) we introduced the synonym generative (general) stochastic network (1) and discriminative (general) stochastic network (2).

In order to fully pass information starting from the bottom layer to the top layer and vice-versa in a GSN, 2*d walk-back steps are needed. We restricted the network to be able to access all information, therefore the setup k=2*d is mandatory.